# Comparing Inertial Measurement Units to Markerless Video Analysis for Movement Symmetry in Quarter Horses

**DOI:** 10.3390/s23208414

**Published:** 2023-10-12

**Authors:** Thilo Pfau, Kiki Landsbergen, Brittany L. Davis, Olivia Kenny, Nicole Kernot, Nina Rochard, Marion Porte-Proust, Holly Sparks, Yuji Takahashi, Kasara Toth, W. Michael Scott

**Affiliations:** 1Faculty of Kinesiology, University of Calgary, Calgary, AB T2N 1N4, Canada; 2Faculty of Veterinary Medicine, University of Calgary, Calgary, AB T2N 1N4, Canadawmichael.scott@ucalgary.ca (W.M.S.); 3Faculty of Biomedical Engineering, University of Calgary, Calgary, AB T2N 1N4, Canada; 4School of Agricultural, Environmental and Veterinary Sciences, Charles Sturt University, North Wagga, NSW 2650, Australia; 5Ecole Nationale Vétérinaire de Toulouse, 31300 Toulouse, France; 6Japan Racing Association, Tokyo 105-0003, Japan

**Keywords:** horse, movement symmetry, inertial measurement unit, markerless tracking, artificial intelligence, lameness

## Abstract

Background: With an increasing number of systems for quantifying lameness-related movement asymmetry, between-system comparisons under non-laboratory conditions are important for multi-centre or referral-level studies. This study compares an artificial intelligence video app to a validated inertial measurement unit (IMU) gait analysis system in a specific group of horses. Methods: Twenty-two reining Quarter horses were equipped with nine body-mounted IMUs while being videoed with a smartphone app. Both systems quantified head and pelvic movement symmetry during in-hand trot (hard/soft ground) and on the lunge (left/right rein, soft ground). Proportional limits of agreement (pLoA) were established. Results: Widths of pLoA were larger for head movement (29% to 50% in-hand; 22% to 38% on lunge) than for pelvic movement (13% to 24% in-hand; 14% to 24% on lunge). Conclusion: The between-system pLoAs exceed current “lameness thresholds” aimed at identifying the affected limb(s) in lame horses. They also exceed published limits of agreement for stride-matched data but are similar to repeatability values and “lameness thresholds” from “non-lame” horses. This is encouraging for multi-centre studies and referral-level veterinary practice. The narrower pLoA values for pelvic movement asymmetry are particularly encouraging, given the difficulty of grading hind limb lameness “by eye”.

## 1. Introduction

Lameness is a frequently encountered problem amongst horses of all disciplines. Quantitative assessment might be the key to improving detection, resolution and follow-up. “Head nod” [1] and “hip hike” [2] are two important signs used for visual detection of lameness and can be expressed by quantifying movement asymmetries between the two halves of a trot stride cycle. Typically, vertical movements of the head (poll) and pelvis (tuber sacrale) are used to measure movement asymmetry as differences between the two displacement minima, reached during the stance phase, or between the two maxima, reached in the aerial phase between stance phases, or between the upward movement amplitudes [3,4].

Movement symmetry analysis can be achieved with body-mounted IMUs (inertial measurement units) [5] or optical methods with cameras [6,7]. Most recently, an artificial intelligence (AI)-based markerless smartphone app (Sleip^®^, Stockholm, Sweden) has been validated against an established marker-based 3D optical motion capture system [8]. The markerless smartphone approach appears to be particularly interesting for clinical practice as it requires minimal infrastructure, only requiring internet access and an optional tripod. In addition to the validated straight-line locomotion [8], the app also allows for the analysis of data from other clinically relevant “in-field” exercises such as on the lunge (i.e., horses moving in circles) on different surfaces.

With an increasing diversity of camera- and IMU-based systems available [5,9,10,11,12], the question of “comparability” is of growing importance. In particular, systems with minimal infrastructure requirements are opening the door for larger-scale and/or longitudinal follow-up studies. Differences between the different systems in sampling rates [13], processing and filtering mechanisms [3,9,12,14,15], as well as stride segmentation [9,16,17,18,19], introduce challenges for direct comparison. Yet, comparison without exact “stride matching” gives a more realistic assessment of how well different systems agree, since the stride selection is typically performed (semi-) automatically with limited possibility for intervention by the user.

Thus, the aim of this study was to conduct an “in-field” comparison of two validated movement symmetry measurement systems, a multi-sensor IMU system [3,4] and a smartphone app [8] for markerless video analysis. The objective of the study was to investigate the limits of agreement (LoA) [20,21] for movement symmetry measures without exact stride matching for a realistic estimation of between-system agreement. Based on the published results of repeatability studies [7,22,23], we hypothesized that non-stride-matched conditions would exceed previously published LoA values calculated in comparison with 3D optical motion capture [3,4,8]. In addition, we were also interested in investigating the possible influence of circular movement (on the lunge) for LoA values.

## 2. Materials and Methods

This study was approved by the University of Calgary Veterinary Sciences Animal Care and Use Committee (approval number AC21-0231).

### 2.1. Horses

Twenty-two reining Quarter horses from two different locations (one private farm and one boarding facility near Calgary, Alberta) with a median (min–max) age of 5 (3–23) years, a body mass of 427.5 (323–532) kg and a height of 14.2 (13.2–15.2) hands were used in the study. Ten mares, ten geldings and two stallions were recruited to this study and were assessed by a board-certified (Dipl. ACVS, Dipl. ACVSMR) veterinarian for their ability to participate in the exercises required by the study protocol: approximately 25 strides in walking and trotting in-hand (hard and soft ground) and on the lunge (soft ground only) in circles. The presence of mild lameness was not considered an exclusion criterion. Horses with obvious lameness when walking, corresponding to AAEP scale grade 4 or higher, were not eligible for inclusion in the study.

### 2.2. Instrumentation and Setup

Each horse was instrumented with a 9-sensor inertial measurement unit (IMU) gait analysis system (Xsens MTw, 47 mm × 30 mm × 13 mm, 16 g) with sensors attached to the following anatomical locations: poll, withers, thoracic vertebra 13 and 18, third lumbar vertebra, between the tubera sacrale, caudal to the sacrum just cranial to the tail root and over the left and right tuber coxae. A tripod (K&F Concept 238 cm with 360 Degree Ball Head SA254T1) with a smartphone holder (Metal Phone Tripod Mount+Rotating Cold Shoe, YISILIN) was set up with the phone (iPhone14Pro) at approximately 1.6 m height and with the phone running the AI-based video analysis app (Sleip) recording 4k video (2160 × 3840 pixels) at a 60 Hz frame rate [8].

### 2.3. Exercise and Data Collection

Horses were led in-hand over a hard surface in walk then trot, followed by in-hand exercise in walk and trot on a soft surface (location 1: reining purposed outdoor arena; location 2: reining purposed indoor arena). Finally, horses were lunged in trot (with head collar and slack lead rope) on the soft surface on the left and then the right rein. The length of the straight-line trotup varied between locations and between surfaces typically less than 30m. Only trot data were analysed in this study. IMU data were transmitted (collection started and stopped manually) via proprietary protocols (Xsens Awinda) at a data rate of 60 Hz (Pfau and Reilly 2020) per individual data channel (tri-axial acceleration (±160 m s^−2^), tri-axial rate of turn (±2000 degree s^−1^) and tri-axial magnetic field strength (±1.9 Gauss) to a laptop computer with an Awinda transceiver (Xsens) and running MTManager (v2020.0.2, Xsens, Enschede, The Netherlands) software. In parallel to IMU data collection, video data were collected, manually starting and stopping video collection via the Sleip smartphone app (v1.3.1, Sleip, Stockholm, Sweden).

### 2.4. Movement Symmetry Parameters

IMU data from poll and tuber sacrale sensors were analysed based on published protocols [3,4] to calculate vertical displacement (in mm). The remaining seven sensors were not analysed for the purpose of the present manuscript. Continuous data streams were then segmented into individual stride cycles [17], and the following average movement symmetry measures were quantified: difference between minima (MnD), between maxima (MxD) and between upward movement amplitudes (UpD) between the two stride halves. The overall vertical movement amplitude (range of motion, ROM, overall maximum–overall minimum) was used for the data normalization of the MnD, MxD and UpD values: MnDnorm = MnD/ROM, MxDnorm = MxD/ROM and UpDnorm = UpD/ROM.

Video data were analysed with the Sleip app, and MnD (Sleip: MinDiff) and MxD (Sleip: MaxDiff) values were noted from the “analysis” table. UpD (not reported in the app) was calculated as the sum of MnD and MxD corresponding to the sign convention used in the Sleip app.

### 2.5. Asymmetry Direction and Data Normalization

The following sign convention was used across the IMU and video analysis: negative values for movement asymmetries related to patterns typically seen in left fore- or hindlimb lame horses; positive values for movement asymmetries related to patterns typically seen in right fore- or hindlimb lame horses.

For comparison reasons, normalized asymmetry parameters were calculated (see the above equations for IMU). Sleip values had to be multiplied by 0.40 for head movement and 0.25 for pelvic movement. These two distinct scaling factors were necessary due to the internal Sleip normalization, where for head movement, patterns with 40% (0.40) asymmetry are assigned a value of 1, while for pelvic motion, 25% (0.25) asymmetry corresponds to a value of 1.

### 2.6. Statistical Analysis

Scatter plots [20] (*x*-axis: average of normalized IMU and video values; *y*-axis: difference between normalized IMU and video values) and LoA values [20] were calculated based on stride averages using the mean and the standard deviation of the difference values between IMU and video data. When scatter plots indicated significant proportional bias, i.e., increasing or decreasing between-system differences with increasing average values, corrected proportional limits of agreement (pLoA) taking into account the proportional bias were calculated following the published regression method [21]. The significance of the existence of proportional bias was assessed by quantifying 95% confidence intervals of the slope of the regression line (at *p* < 0.05).

In addition to scatter plots displaying the differences between measurements (*y*-axis) over the average measurements (*x*-axis) [20], quantile–quantile (QQ) plots were created (MATLAB) for quantiles of measurement differences (*x*-axis) against quantiles of average measurements in order to illustrate systematic differences that may affect particular values, for example, extreme values [24].

## 3. Results

A total of 2055 strides were recorded with the IMUs for the 22 Quarter horses with an average of 23.35 strides per horse and condition, with the minimum being 7 and the maximum 47, and a standard deviation of 7.79. A total of 2324 strides were recorded with the video app for head movement with an average of 26.71 strides per horse and condition and 1669 for pelvic movement with an average of 19.41 strides per horse and condition. The minima were 10 for the front and 7 for the hind; the maximum was 52 for both. Standard deviations were 9.27 for the front and 8.70 for the hind. A breakdown of the number of strides collected for different exercise conditions for the two systems can be found in Table 1.

Table 2 and Table 3 present the recorded movement symmetry values for the 22 reining Quarter horses included in the study. Values are given for means and standard deviations (SDs) based on the implemented normalization procedure (see Section 2). Mean values for both systems are small, with up to 3% magnitude measured across all conditions and up to 7% magnitude for straight-line trot. Except for pelvic MxD for both systems (Table 2 and Table 3) and pelvic UpD for the video-based analysis (Table 3), the left circle averages show negative values and right circle averages positive values. SD values are generally higher for poll movement symmetry (17 to 37% for IMU, 14 to 41% for video) and lower for pelvic movement symmetry (9 to 22% for IMU, 7 to 18% for video).

Appendix A provide a more detailed illustration of the movement symmetry values recorded for the N = 22 reining Quarter horses across the different combinations of movement directions and surfaces. Three groups of horses were assessed according to head and pelvic movement asymmetry patterns. One group of horses showed consistent evidence of left-sided asymmetry (i.e., reduced force production with the left (fore-, respectively hind) limb), one group showed consistent evidence of right-sided asymmetry (i.e., reduced force production with the right (fore-, respectively hind) limb and one group showed inconsistent movement asymmetry across the three head respectively the three pelvic movement asymmetry parameters.

Table 4 and Table 5 present characteristic values for LoA [20] and pLoA [21] for the comparison of normalized movement symmetry values (MxD_norm_, MnD_norm_ and UpD_norm_). Based on the visual assessment of scatter plots of between-system differences over between-system averages (Figure 1 and Figure 2) and the calculated confidence intervals for slope values (Table 4 and Table 5: slope conf.), in addition to LoA widths, pLoA widths are calculated over ±20% movement symmetry intervals.

Table 4 shows values for in-hand trot. For head movement symmetry, pLoA width varies between 29% for poll MxD_norm_ and 50% for poll UpD_norm_. For pelvic movement symmetry, pLoA widths are approximately half the head movement symmetry values and vary between 13% for pelvic MxD_norm_ and 24% for pelvic UpD_norm_.

Table 5 shows values for circular trot. For head movement symmetry, pLoA width varies between 22% for poll MxD_norm_ and 38% for poll UpD_norm_. For pelvic movement symmetry, pLoA widths vary between 14% for pelvic MxD_norm_ and pelvic MnD_norm_ and 23% for pelvic UpD_norm_.

Quantile–quantile plots of straight-line and circular trot for measurement averages (*y*-axis: (EQ + SL)/2) against measurement differences (*x*-axis: EQ-SL) illustrate a general trend for increased difference magnitudes for more extreme average measurement values (Appendix A). This can be appreciated visually by the increasing distances between the individual data points (blue crosses) and the lines of best fit (red lines). Generally, larger deviations are discernible for movement symmetry of the poll, for which larger average values have been recorded.

## 4. Discussion

This study set out to implement a practically relevant comparison of two commercially available methods for quantifying the movement symmetry of the vertical displacement of the head and pelvis in trotting horses. Owing to the challenge that the video-based system provides normalized movement symmetry values [8], presumably in an attempt to “correct for” differences in movement symmetry observed between head and pelvic movement symmetry for horses of similar lameness grades [11], LoA and pLoA values [20] were calculated as normalized values relative to the overall ROM, as calculated using the IMU system. In contrast to 3D optical motion capture systems making use of multiple synchronized cameras that can provide positional data in “real-world” coordinates (millimetres), when using a single 2D camera without calibration objects of known dimensions, the video app provides values that are normalized to the ROM [8]. In addition to the normalization to the ROM, the values presented in the video app are “standardized” as follows: normalized asymmetry values of head movement are “standardized” into a value of 1 for asymmetry values of 0.4 (40% of ROM) and values of pelvic movement are “standardized” into a value of 1 for asymmetry values of 0.25 (25% of ROM), presumably in an attempt to present values of similar magnitudes for comparable lameness “grades”.

When comparing the results of the present study to previously published movement symmetry values, it may be useful to transform the normalized (proportional) LoA values presented here back into millimetres (“real-world” coordinates). Across all exercise conditions, the vertical displacement mean (min–max) ROM was 62.5 (39,105) mm for the head and 74.7 (42,113) mm for the pelvis. In addition to constant bias and LoA, linear fits to scatter plots of between-system average and between-system difference values (Figure 1: straight line, Figure 2: lunge) were calculated. Proportional bias and the according upper and lower LoA were calculated to provide a fairer assessment of between-system agreement [20,21].

Of higher practical relevance than the generally small bias values (Table 4 and Table 5), are the widths of the LoA, which indicate a range of values containing 95% of the between-system differences for the cohort of reining Quarter horses investigated here. This is of practical value when presented with a previously measured value from one of the two systems and comparing it to a new value measured with the other system. If the newly measured value is outside the lower and the upper LoA value, then the new value can confidently be considered different.

Widths of proportional LoA (Table 4 and Table 5) showed larger values for head movement compared to pelvic movement, ranging from 29% to 50% for in-hand and from 22% to 38% for circular trot for head movement and from 13% to 24% for in-hand and from 14% to 24% for circular trot for pelvic movement. This is expected as, for example, repeatability studies have consistently shown that variability is higher for head movement symmetry between strides [7,22,23]. All three studies quantified pelvic movement symmetry to have approximately half the variability, and a specific decomposition method has been described previously, addressing the removal of extraneous head movement patterns [14]. The two systems compared here both implement similar filtering methods based on highpass filtering [3,4,8]. For the average (min–max) stride time of 695 (593, 839) ms, the video-based system uses a 1.08 (1.26, 0.89) Hz highpass filter cutoff, while the IMU-based system implements a fixed 1 Hz cutoff. Based on a previous investigation about the effects of filtering [15], the largest effect of the generally small between-system difference in highpass filter cutoff frequency would be expected for stride-to-stride variability and in particular for head movement, which is naturally less consistent between strides [7,22,23]. The lowpass filter cutoff of 2.42 times stride frequency applied in the video-based system, as part of a bandpass filter, is—for the highest stride frequencies recorded here (1.69 Hz @ 593 ms stride time)—significantly lower than 10 Hz. An investigation into the influence of sample rate [13] identified a step increase in error when dropping the sample rate from 25 Hz to 20 Hz, suggesting that signal frequencies of up to 10 Hz (i.e., Nyquist rate of half the sample rate) and above are essential for the double integration process of IMU data. The considerably lower cutoff used for the video-based system indicates that these components are less important for video data and the associated markerless tracking.

Another topic of interest is how the LoA values compare to “lameness thresholds”. For the purpose of identifying the most likely affected limb in a horse presenting with a lameness, thresholds of 6 mm for head and 3 mm for pelvic movement symmetry have been introduced previously [25] and are meant to be used for the identification of changes after diagnostic analgesia. Based on a method comparison study [26], these values need adjusting to the specific IMU system used here to >8 mm (head) and >4 mm (pelvis). Using the average ROM values of 62.5 mm and 74.7 mm for the head and pelvis, thresholds of −36% to +24% for head movement and −7% to +18% for pelvic movement presented in a recent study for in-hand trot [10] would result in higher mm thresholds of −22 to +15 mm for head movement and −5 to +13 mm for pelvis movement.

The same ROM values used above can be utilized for the calculation of mm values from the % LoA values presented here. The proportional widths of the LoA between the video and IMU system of 29% (±14.5% around the mean) and 39% (±19.5%) for head MxD and MnD then translate into ±9 mm and ±12 mm. The pLoA widths for pelvic movement of 13% (±6.5%) for MxD and 18% (±9%) for MnD translate into ±5 mm and ±7 mm. These “un-normalized” values are between 1.125 and 1.5 higher than the 8 mm threshold [25,26] and between 1.25 and 1.75 higher than the 4 mm threshold [25,26]. Compared to recently published thresholds [12], the LoA values presented here are of similar magnitudes.

Interestingly, a recently published study comparing the output of the video app to stride-matched measurements from an optical motion capture system providing three-dimensional coordinates of markers attached to horses provided considerably narrower LoA values for trial-by-trial comparisons of around ±5 to 7% of ROM [8]. In addition to a stride-matched analysis, that study employed identical filtering [15] and processing methods for the two optical methods—the 2D markerless method used here and one 3D marker-based method. The quantile–quantile plots of the present between-system comparison with averages plotted against between-system differences (Appendix A) indicate that there may be a tendency for systematical differences for more extreme values representing exercises with increased movement asymmetry (typically more severe lameness, e.g., [10]). The origin of these differences is currently not clearly understood. One possible explanation might be the change in vertical displacement from a reduced movement amplitude during the lame limb stance phase in mildly lame horses [1] to a complete lack of a displacement peak during the lame stance phase for moderately to severely lame horses [27]. Depending on what exact methods are being used for detecting displacement minima and maxima, systematic differences might arise, possibly in relation to alterations in the timing of head excursion relative to trunk movement [28].

Another contributing factor to between-system differences might be the use of circular exercise. Increased movement asymmetries may be found in particular for horses already presenting with mild movement asymmetry on the straight [29,30,31]. Different regression equations for correcting systematic between-system differences between the IMUs used here and a second IMU-based system have been suggested in previous work [26]. To the authors’ knowledge, no comparison study has been published for the video-based system for circular exercise. However, given that quantile–quantile plots for both straight-line (Appendix A) and circular trot (Appendix A) show systematic differences, lunge exercise is unlikely to be the only contributing factor. The results of the present study with similar widths of limits of agreement for straight line and circular exercise (Table 4 and Table 5) suggest that circular exercise can be assessed without any negative consequences for between-system precision.

It may also be interesting to further investigate specific factors in association with the accuracy and precision of specific anatomical landmarks on the bodies of horses. The published validation study of the video app [8] does not provide specific details about the identified landmarks or the data base used for model training. These are factors that are of general interest for pose estimation in animals with a freely available deep learning approach [32] as well as for its application in automated lameness detection in horses [33]. Without further knowledge of the training conditions and classification approaches, the current study can only provide an estimate of the limits of agreement achieved under the specific conditions encountered here at the two different data collection locations.

This study was conducted in a convenience sample of N = 22 Quarter horses differing in age and sex and with varying levels of experience in reining. Horses had to be able to perform in-hand, walk and trot exercise as well as lunge exercise in trot on a reining-purposed arena surface. They did not have to be completely free from lameness, and the lameness grade and each horse’s ability to perform the required exercises was judged by a qualified, board-certified veterinary surgeon (WMS) who was present for the whole data collection period. When calculating LoA values, in particular for data with proportional bias, it is desirable that the between-system average values cover a wide range. This would not have been the case to the same degree when restricting data collection to non-lame horses and when only conducting measurements during straight-line trot. Given the diverse characteristics of the included Quarter horses, it may be necessary to conduct further studies including horses with more narrowly defined characteristics in order to investigate how representative our system comparison results are.

## 5. Conclusions

The proportional limits of agreement between the IMU- and video-based systems used here exceed established “lameness thresholds” for identifying the most likely affected limb in clinically lame horses and are wider than limits of agreement for stride-matched data for a comparison of the video-based system to three-dimensional optical motion capture. However, the agreement is similar to repeatability studies for IMU- and motion capture-based studies [7,22,23] and compared to thresholds derived from “non-lame” horses [12]. This is very encouraging for conducting larger-scale, multi-centre studies with the video- and IMU-based approaches compared here including straight-line, in-hand trot as well as circular lunge exercise, for which similar limits of agreement were found. The narrower limits of agreement for pelvic movement asymmetry are of particular interest, given the difficulty of grading hind limb lameness “by eye”. Due to the study design here, aiming at comparing the two techniques but not treating one or the other method (or the expert visual assessment) as the “gold standard”, it has not been possible to quantify the sensitivity or specificity of either of the approaches.

## Figures and Tables

**Figure 1 sensors-23-08414-f001:**
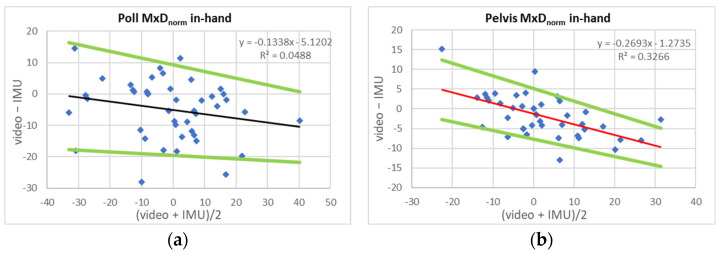
Scatter plots and regression-based corrected proportional limits of agreement (proportional bias: black/red line; lower and upper limits of agreement: green lines, see Table 4 for equations, [21]) between IMU- and video-based measurements for three movement symmetry measures (MxDnorm, MnDnorm and UpDnorm) for poll and pelvis for N = 22 Quarter horses trotting **in-hand**. **Red line**: slope confidence interval does not include zero. (**a**) Poll MxD_norm_, (**b**) pelvis MxD_norm_, (**c**) poll MnD_norm_, (**d**) pelvis MnD_norm_, (**e**) poll UpD_norm_, (**f**) pelvis UpD_norm_. All values in % of range of motion.

**Figure 2 sensors-23-08414-f002:**
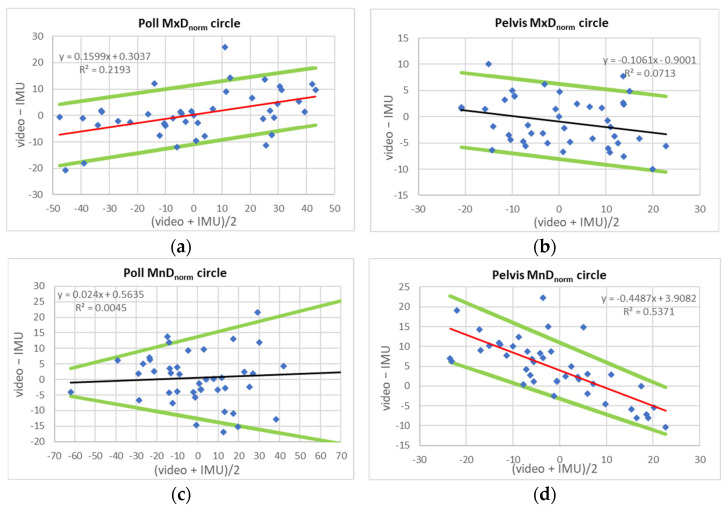
Scatter plots and regression-based corrected proportional limits of agreement (bias: black/red line; lower and upper limits of agreement: green lines; see Table 5 for equations) between IMU- and video-based measurements for three movement symmetry measures (MxDnorm, MnDnorm and UpDnorm) for poll and pelvis for N = 22 Quarter horses trotting on the lunge (left and right rein). **Red line**: slope confidence interval does not include zero. (**a**) Poll MxD_norm_, (**b**) pelvis MxD_norm_, (**c**) poll MnD_norm_, (**d**) pelvis MnD_norm_, (**e**) poll UpD_norm_, (**f**) pelvis UpD_norm_. All values in % of range of motion.

**Table 1 sensors-23-08414-t001:** Number of strides for IMU- (EquiGait) and video-based (Sleip) analysis in different conditions: in-hand: straight line trot on soft and hard ground; lunge left/lunge right: right or left circle trot on soft ground; hard: hard ground on straight; soft: soft ground across straight line, left circle and right circle.

**IMU (EquiGait)**											
	In-hand	Lunge left		Lunge right		Hard			Soft		
**avg**	18.68	29.57		26.65		17.2			25.61		
**SD**	6.87	6.16		4.72		5.32			7.38		
**Video (Sleip)**											
	Front					Hind					
	In-hand	Lunge left	Lunge right	Hard	Soft	In-hand	Lunge left	Lunge right		Hard	Soft
**avg**	21.29	31.80	32.1	21.5	28.60	13.74	24.60	25.5		13.35	21.56
**SD**	5.39	11.16	7.66	4.97	9.81	5.28	8.52	7.42		4.60	9.05

**Table 2 sensors-23-08414-t002:** Averages and standard deviations of **IMU-based** head and pelvis movement asymmetries for normalized values (in percentage of ROM) for N = 22 reining Quarter horses. In-hand: straight line trot on soft and hard ground; lunge left/lunge right: right or left circle trot on soft ground; hard: hard ground on straight; soft: soft ground on straight, left circle and right circle. Multiply normalized value (%) by ROM (mm) and divide by 100 to obtain asymmetry in mm.

	**Poll**							
	**MxD (%)**		**MnD (%)**		**UpD (%)**		**ROM (mm)**	
	**Mean**	**SD**	**Mean**	**SD**	**Mean**	**SD**	**Mean**	**SD**
Total	1.25	20.42	−0.48	22.00	1.20	36.04	62.48	11.73
In-hand	1.43	17.07	1.21	19.80	0.40	29.25	59	10.16
Left	−10.45	20.52	−12.60	21.09	−21.40	33.65	65.5	12.87
Right	12.60	19.97	13.09	19.31	25.39	35.39	66.41	11.38
Hard	1.65	17.98	1.69	21.20	4.31	32.40	54.68	9.47
Soft	1.12	21.17	−1.20	22.20	0.16	37.11	65.07	11.25
	**Pelvis**							
	**MxD (%)**		**MnD (%)**		**UpD (%)**		**ROM (mm)**	
	**Mean**	**SD**	**Mean**	**SD**	**Mean**	**SD**	**Mean**	**SD**
Total	2.18	12.60	−3.37	13.51	−1.54	19.34	74.70	12.31
In-hand	2.76	12.62	−3.06	12.03	−1.04	21.44	68.77	10.27
Left	8.24	9.47	−12.47	8.93	−4.23	12.99	80.45	12.46
Right	−5.05	11.74	5.09	14.37	0.14	19.95	80.82	9.96
Hard	3.48	11.84	−3.76	12.44	1.15	21.61	62.91	8.38
Soft	1.75	12.82	−3.25	13.84	−2.44	13.44	78.64	10.81

**Table 3 sensors-23-08414-t003:** Averages and standard deviations of **video-based** head and pelvis movement asymmetries for normalized values (in percentage of ROM) for N = 22 reining Quarter horses. Total: across all exercise conditions; in-hand: straight line trot on soft and hard ground; lunge left/lunge right: right or left circle trot on soft ground; hard: hard ground on straight; soft: soft ground on straight, left circle and right circle. Multiply normalized value (%) by ROM (mm, Table 2) and divide by 100 to obtain asymmetry in mm.

	**Poll**					
	**MxD (%)**		**MnD (%)**		**UpD (%)**	
	**Mean**	**SD**	**Mean**	**SD**	**Mean**	**SD**
Total	−0.96	22.17	−0.87	21.19	−1.84	37.97
In-hand	−3.54	15.09	−3.09	17.37	−6.64	27.05
Left	−13.71	23.34	−12.00	18.55	−25.71	32.32
Right	16.36	22.49	14.18	22.39	30.54	40.60
Hard	−4.73	14.30	−1.45	19.95	−6.18	29.48
Soft	0.31	24.10	−0.68	21.59	−0.37	40.29
	**Pelvis**					
	**MxD (%)**		**MnD (%)**		**UpD (%)**	
	**Mean**	**SD**	**Mean**	**SD**	**Mean**	**SD**
Total	0.64	10.38	1.02	8.59	1.66	14.33
In-hand	1.10	9.69	0.99	7.56	2.09	15.35
Left	6.43	8.35	−3.69	7.33	2.74	9.98
Right	−5.79	10.09	5.57	9.26	0.23	15.66
Hard	2.04	10.41	1.93	9.17	3.98	17.77
Soft	0.16	10.33	0.70	8.37	0.86	12.88

**Table 4 sensors-23-08414-t004:** Limits of agreement (LoA) between IMU- and video-based assessment of three movement symmetry variables (MnDnorm, MxDnorm, UpDl) for head (poll) and pelvis calculated for **in-hand trot** in N = 22 Quarter horses. Given are constant bias, upper and lower LoA (uLoA and lLoA) and width of LoA (LoA width). Also given are proportional limits of agreement (pLoA) based on regression equations [21] for all variables: variables x: (video + IMU)/2; y: (video—IMU). Prop. Bias: equation for the proportional bias (red/black line, Figure 1); slope conf.: confidence interval of the slope of proportional bias. Values in red indicate that the slope confidence interval does not include zero. Regression lines are then displayed in red (Figure 1). uLoA, lLoA: lower and upper limit of agreement. pLoA width over (±20%): width of proportional limits of agreement calculated over a range of ±20% asymmetry.

**Poll**	**MxD_norm_**	**MnD_norm_**	**UpD_norm_**
	**constant bias**		
const. bias	−4.98 [−7.9; −2.1]	−1.88 [−5.6; 1.0]	−7.04 [−12.1; −2.0]
uLoA	13.54 [8.6; 18.5]	21.92 [15.5; 28.3]	25.17 [16.5; 33.8]
lLoA	−23.50 [−28.5; −18.5]	−25.69 [−32.1; −19.3]	−39.25 [−47.9; −30.6]
LoA width	37.04	47.61	64.42
	**proportional bias**		
prop. Bias	y = −0.1338x − 5.1202	y = −0.1456x − 2.198	y = −0.0854x − 7.3073
slope conf.	[−0.32; 0.05]	[−0.35; 0.06]	[−0.27; 0.10]
uLoA/lLoA	y = −0.1338x − 5.1202 ± 1.96 (−0.0406x + 7.3503)	y = −0.1456x − 2.198 ± 1.96 (0.1248x + 8.6149)	y = −0.0854x − 7.3073 ± 1.96 (−0.0495x + 12.805
pLoA width	28.81	38.77	50.19

**Pelvis**	**MxD_norm_**	**MnD_norm_**	**UpD_norm_**
	**constant bias**		
const. bias	NA	NA	NA
uLoA	NA	NA	NA
lLoA	NA	NA	NA
LoA width	NA	NA	NA
	**proportional bias**		
prop. Bias	y = −0.2693x − 1.2735	y = −0.3665x + 4.3923	y = −0.3104x + 2.5489
slope conf.	**[−0.39; −0.15]**	**[−0.58; −0.15]**	**[−0.46; −0.16]**
uLoA/lLoA	y = −0.2693x − 1.2735 ± 1.96 (−0.026x + 3.266)	y = −0.3665x + 4.3923 ± 1.96 (0.0532x + 4.465)	y = −0.3104x + 2.5489 ± 1.96 (0.0689x + 6.0484)
pLoA width	12.80	17.50	23.71

**Table 5 sensors-23-08414-t005:** Limits of agreement (LoA) between IMU- and video-based assessment of six movement symmetry variables (MnDnorm, MxDnorm, UpDnorm) for head (poll) and pelvis calculated for **circular trot** (left and right rein) in N = 22 Quarter horses. Given are constant bias, upper and lower LoA (uLoA and lLoA) and width of LoA (LoA width). Also given are proportional limits of agreement (pLoA) based on regression equations (Bland and Altman, 1999) for all variables: variables x: (video + IMU)/2; y: (video—IMU). Prop. Bias: equation for the proportional bias (red/black line, Figure 2); slope conf.: confidence interval of the slope of proportional bias. Values in red indicate that the slope confidence interval does not include zero. Regression lines are then displayed in red (Figure 2). uLoA, lLoA: lower and upper limit of agreement. pLoA width over (±20%): width of limits of agreement calculated over a range of ±20% asymmetry.

**Poll**	**MxD_norm_**	**MnD_norm_**	**UpD_norm_**
	**constant bias**		
const. bias	NA	0.52 [−2.12; 3.17]	NA
uLoA	NA	17.38 [12.86; 21.90]	NA
lLoA	NA	−16.34 [−20.86; −11.81]	NA
LoA width	NA	33.72	NA
	**proportional bias**		
prop. Bias	y = 0.1599x + 0.3037	y = 0.024x + 0.5635	y = 0.0977x + 0.5319
slope conf.	**[0.04; 0.18]**	[−0.04; 0.11]	**[0.02; 0.15]**
lLoA/uLoA	y = 0.1599x + 0.3037 ± 1.96 (−0.0043x + 5.7197)	y = 0.024x + 0.5635 ± 1.96 (0.0717x + 6.7051)	y = 0.0977x + 0.5319 ± 1.96 (0.0352x + 9.6432)
pLoA width	22.42	26.28	37.80

**Pelvis**	**MxD_norm_**	**MnD_norm_**	**UpD_norm_**
	**constant bias**		
const. bias	−0.86 [−2.29; 0.57]	NA	NA
uLoA	8.24 [5.80; 10.69]	NA	NA
lLoA	−9.96 [−12.40; −7.52]	NA	NA
LoA width	18.20	NA	NA
	**proportional bias**		
prop. Bias	y = −0.1061x − 0.9001	y = −0.4487x + 3.9082	y = −0.2653x + 3.3081
slope conf.	[−0.23; 0.01]	**[−0.58; −0.32]**	**[−0.44; −0.09]**
lLoA/uLoA	y = −0.1061x − 0.9001 ± 1.96 (0.0016x + 3.6528)	y = −0.4487x + 3.9082 ± 1.96 (−0.0266x + 3.6142)	y = −0.2653x + 3.3081 ± 1.96 (0.0257x + 5.956)
pLoA width	14.32	14.17	23.35

## Data Availability

Data are available at 10.6084/m9.figshare.24150894.

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
