# Peer review of "Comparing Inertial Measurement Units to Markerless Video Analysis for Movement Symmetry in Quarter Horses"

_sensors, 2023, doi:10.3390/s23208414_

Round 1
Reviewer 1 Report
The article “Comparing inertial measurement units to markerless video 2 analysis for movement symmetry in Quarter horses” compares the symmetry measurements of horse locomotion obtained from an IMU system with a markerless IA based system.
The paper is well written and structured; The purpose of the study is clearly explained and the authors propose an adapted experimental protocol.
However, before acceptance for publication, some additions and clarifications should be made:
1.The linear regression to determine the proportional bias has values of the coefficient of determination (R2) less than 0.5371, which does not guarantee that the linear regression is representative. It is suggested here either not to consider this "proportional bias" at first instance, or to do additional statistical tests to demonstrate that linear regression has a meaning.
2The authors suggest that the observed quantitative difference between IMU and unlabeled system measurements may be related to cut-off frequency. This could be possible ; but i think it's mostly related to the detection method that the markerless system uses. Indeed, this markerless system, as used here, is based on the detection of landmark on the horse by means of deep learning network previously trained with a database. However, the only one article referenced [8 ]which presents the technique remains very vague and incomplete on image processing and deep learning techniques. In particular, in addition to the detection of a marker, a marker tracking is conventionally associated by interpolation and/or a prediction of the displacement;. Moreover, since no details are given on the given training database used for deep learning network, it is difficult to know whether measurement in a real environment such as the authors do is taken into account in the given training database. This long commentary highlights above all the non-transparency of the so-called IA systems. It is suggested here that the authors should focus the discussion on this non-transparency theme, which does not anticipate the errors or intrinsic limitations of the IA-based video analysis app.
The theme of this article is the comparison between two systems; but it's also a metrology problem. As a result, the question of accuracy and precision still arises. Indeed, a system is expected to do the "right" measurement to make the "right" diagnosis. In clinical terms, this amounts to assessing specificity and sensitivity; although this point cannot be made in this study (because of homogeneous population or low lameness) here, this can be presented as a limitation of the current study.
Specific remarks
Line 89 Specify the image size and frequency
Line 123-127: the need to correct markerless values should be clarified.
Author Response
The article “Comparing inertial measurement units to markerless video 2 analysis for movement symmetry in Quarter horses” compares the symmetry measurements of horse locomotion obtained from an IMU system with a markerless IA based system.
The paper is well written and structured; The purpose of the study is clearly explained and the authors propose an adapted experimental protocol.
Thank you for your positive assessment.
However, before acceptance for publication, some additions and clarifications should be made:
1.The linear regression to determine the proportional bias has values of the coefficient of determination (R2) less than 0.5371, which does not guarantee that the linear regression is representative. It is suggested here either not to consider this "proportional bias" at first instance, or to do additional statistical tests to demonstrate that linear regression has a meaning.
We sincerely apologize for being unclear on this point and for having made a mistake in some of the sub-panels of figure 2 in which ALL of the regression lines were ‘red’ indicating a slope value significantly different from zero. We have modified the manuscript in multiple places:
- We have clarified in the materials and methods section that significance of the slope values has been assessed:
“When scatter plots indicated significant proportional bias, i.e. increasing or decreasing between-system differences with increasing average values, corrected proportional limits of agreement (pLoA) taking into account the proportional bias were calculated following the published regression method [21]. Significance of the existence of proportional bias was assessed through quantifying 95% confidence intervals of the slope of the regression line at P<0.05. - Data tables: in tables 4 and 5, slope values for which the 95% confidence intervals do not include zero (i.e. indicate a slope significantly different from zero at P<0.05) are now given in ‘red’ corresponding to the color scheme used for the regression lines in the lines in the Bland and Altman style plots (red for regression lines with slope significantly different form zero, black for lines with slope not significantly different from zero according to the confidence interval).
- Revised figures 1 and 2 now correspond in ‘color scheme’ to the significance of the slope reported in tables 4 and 5. In particular, figure 2, subpanels for poll MnD and pelvic MxD now show ‘black’ regression lines which previously erroneously were given in ‘red’ color.
2 The authors suggest that the observed quantitative difference between IMU and unlabeled system measurements may be related to cut-off frequency. This could be possible ; but i think it's mostly related to the detection method that the markerless system uses. Indeed, this markerless system, as used here, is based on the detection of landmark on the horse by means of deep learning network previously trained with a database. However, the only one article referenced [8 ]which presents the technique remains very vague and incomplete on image processing and deep learning techniques. In particular, in addition to the detection of a marker, a marker tracking is conventionally associated by interpolation and/or a prediction of the displacement;. Moreover, since no details are given on the given training database used for deep learning network, it is difficult to know whether measurement in a real environment such as the authors do is taken into account in the given training database. This long commentary highlights above all the non-transparency of the so-called IA systems. It is suggested here that the authors should focus the discussion on this non-transparency theme, which does not anticipate the errors or intrinsic limitations of the IA-based video analysis app.
Thank you for this insightful comment. We have amended the discussion to the best of our ability. It is somewhat difficult given that, as the reviewer also states above, the information about the smartphone app is somewhat limited with respect to multiple components that may or may not be relevant here including the training set and the landmarks used for markerless tracking. We have added the below paragraph to the discussion:
“It may also be interesting to further investigate specific factors in association with accuracy and precision of specific anatomical landmarks on the body of the horses. The published validation study of the video-app [8] does not provide specific details about the identified landmarks or the database used for model training. These are factors that are of general interest for pose estimation in animals with a freely available deep learning approach [32] as well as for application in automated lameness detection in horses [33]. Without further knowledge of the training conditions and classification approaches, the current study can only provide an estimate of the limits of agreement achieved under the specific conditions encountered here at the two different data collection locations.”
The theme of this article is the comparison between two systems; but it's also a metrology problem. As a result, the question of accuracy and precision still arises. Indeed, a system is expected to do the "right" measurement to make the "right" diagnosis. In clinical terms, this amounts to assessing specificity and sensitivity; although this point cannot be made in this study (because of homogeneous population or low lameness) here, this can be presented as a limitation of the current study.
Thank you for this suggestion. We have added the following sentence to the conclusion section:
“Due to the study design here, aiming at comparing the two techniques but not treating one or the other method (or the expert visual assessment) as the ‘gold standard’, it has not been possible to quantify sensitivity or specificity of either of the approaches.”
Specific remarks
Line 89 Specify the image size and frequency
We have added information about image size and frame rate from the published validation study of the video-app.
Line 123-127: the need to correct markerless values should be clarified.
Thank you for this comment. We have made an addition to the relevant discussion section and indicate that this normalization is likely in relation to different amounts of asymmetry present between head and pelvic movement for similar degrees of lameness (including a reference to a published study presenting relevant information about this topic).
Reviewer 2 Report
Very interesting data showing a comparison of two diagnostic methods for lameness in horses. I believe that minor revisions with some edits/suggestions for further consideration could enhance the paper.

Author Response
Very interesting data showing a comparison of two diagnostic methods for lameness in horses. I believe that minor revisions with some edits/suggestions for further consideration could enhance the paper.
Thank you for the positive comments.
Line 54: Is this the problem of the research? I suggest you remove this question from the introduction. If it is the research question, move it to the end of the introduction.
Thank you. We have removed this sentence.
Line 61: I suggest changing the term “real world” for “field tests”.
Thank you. We have replaced the term ‘real world’ with ‘in-field’ where referring to the conducting exercises in the ‘field’ (i.e. not in the lab) but have chosen to still use the term ‘real world’ when referring to measurements expressed in millimeters.
Line 80: Describe the exclusion criteria. How was the presence or absence of lameness defined? On what scale? It would be interesting to include in the work how many animals were lame, the degree, and the results of the comparison between the two methods in healthy and lame animals.
Thank you for highlighting this. We have further clarified that horses which showed a lameness at the walk (corresponding to a lameness grade of 4 or above on the AAEP scale) were not eligible for inclusion into the study. In addition, we have added two additional figures (Figures S1 and S2 in the revised version) that show in more detail the distribution of head and pelvic movement symmetry parameters of the horses included in the study. This illustrates to the informed reader the spread of movement asymmetries that are observable across horses under the different exercise conditions. Given the variability in visual lameness scoring between observers and given that movement symmetry measurements are available for the horses under multiple conditions, we have instead chosen to provide additional detailed quantitative data in the form of supplementary figures. These may prove useful for future overview or meta-studies in terms of discussing the nature of the horses included in individual studies.
The secondary objective: “In addition, we were also interested in investigating the possible influence of circular movement (on the lunge) for LoA values.” was not discussed and there is no conclusion of this. I suggest including the discussion of this topic and concluding something about it.
Thank you again for highlighting that we had failed to adequately conclude on this topic in the discussion. We have added an explicit conclusion sentence to the paragraph which addresses the topic:
“Another contributing factor to between-system differences might be the use of circular exercise. Increased movement asymmetries may be found in particular for horses already presenting with mild movement asymmetry on the straight [29–31]. Different regression equations for correcting systematic between-system differences between the IMUs used here and a second IMU-based system have been suggested in previous work [26]. To the authors’ knowledge, no comparison study has been published for the video-based system for circular exercise. However, given that quantile-quantile plots for both straight-line (Figure S3) and circular trot (Figure S4) show systematic differences, lunge exercise is unlikely the only contributing factor. The results of the present study, with similar widths of limits of agreement for straight line and circular exercise (Tables 4 and 5), suggest that circular exercise can be assessed without any negative consequences for between-system precision.
We have also modified the relevant part of the conclusion as follows:
“This is very encouraging for conducting larger scale, multi-centre studies with the video and IMU based approaches compared here including straight-line, in-hand trot as well as circular lunge exercise for which similar limits of agreement were found.”
Round 2
Reviewer 1 Report
The authors answer the questions well. The study can be published.